# Monocytes and Macrophages in Kidney Disease and Homeostasis

**DOI:** 10.3390/ijms25073763

**Published:** 2024-03-28

**Authors:** Rajesh Nachiappa Ganesh, Gabriela Garcia, Luan Truong

**Affiliations:** 1Department of Pathology and Genomic Medicine, Houston Methodist Hospital, Houston, TX 77030, USA; ltruong@houstonmethodist.org; 2Department of Pathology, Jawaharlal Institute of Postgraduate Medical Education and Research, Puducherry 605006, India; 3Department of Medicine, Renal Division, University of Colorado, Anschutz Medical Campus, Aurora, CO 605006, USA; gabriela.garcia@cuanschutz.edu

**Keywords:** monocytes, macrophages, dendritic reticulum cells, M1 and M2 phenotypes, glomerulonephritis, tubulointerstitial nephritis, interstitial fibrosis, tubular atrophy, hypertension

## Abstract

The monocyte–macrophage lineage of inflammatory cells is characterized by significant morphologic and functional plasticity. Macrophages have broad M1 and M2 phenotype subgroups with distinctive functions and dual reno-toxic and reno-protective effects. Macrophages are a major contributor to injury in immune-complex-mediated, as well as pauci-immune, glomerulonephritis. Macrophages are also implicated in tubulointerstitial and vascular disease, though there have not been many human studies. Patrolling monocytes in the intravascular compartment have been reported in auto-immune injury in the renal parenchyma, manifesting as acute kidney injury. Insights into the pathogenetic roles of macrophages in renal disease suggest potentially novel therapeutic and prognostic biomarkers and targeted therapy. This review provides a concise overview of the macrophage-induced pathogenetic mechanism as a background for the latest findings about macrophages’ roles in different renal compartments and common renal diseases.

## 1. Introduction

Mononuclear phagocytic systems include monocytes, macrophages, and dendritic cells. These are all categorized as cells of myeloid lineage arising from bone marrow precursors, with subsequent differentiation based on injurious stimuli. Monocytes are one type of inflammatory cell that account for about 2% to 8% of leukocytes in circulation during the physiologic homeostatic state. In pathologic conditions, monocytes exit from the circulation to reside in tissue and are then called “macrophages”. In the tissue, macrophages acquire additional genetic and phenotypic attributes that enable them to affect many functions. During this metamorphosis, a subset of macrophages gains a distinctive phenotype, which includes delicate cytoplasmic processes and the ability to present antigens to the immune-defense machinery. This type of specialized macrophage is known as a dendritic cell. Each of these cell types have unique as well as overlapping functions which are critical in immune homeostasis.

Recent studies demonstrate that a significant population of tissue macrophages are not of monocytic origin, but develop in situ from various resident cells, including embryonic stem cells originating from the yolk sac and from fetal liver monocytes. [1,2]. These tissue-specific macrophages have unique functions depending on the organs or tissue they inhabit. For example, removal of surfactant is a major function of resident tissue-specific macrophages in the lung. In the kidneys, tissue-specific macrophages identify immune deposits by activating the FCγR-IV-dependent response [2]. Macrophages are known to spearhead effective functioning across different organs through several mechanisms that are broadly categorized into different mechanistic domains: self-cytokine-mediated effects; inflammatory cell-to-cell immunomodulatory functions; immunologic initiation and modulation, including antigen-presenting role; and immune-protective functions, including clearance of toxic molecules of immune complexes. These tissue-specific macrophages are distinct from marrow-derived monocytes, which further differentiates the macrophage subtypes [2].

There are distinct subtypes of monocytes, dendritic cells, and macrophages, with specialized though overlapping functions. These subtypes are discussed comprehensively to provide a broad template in this review, along with the major cytokines and chemokines that regulate the transformation of these cells. Knowledge of the etiopathogenetic link and disease mechanisms has evolved based on in vitro cell lines and in vivo studies in experimental animals, followed by study of human diseases. This review focuses on monocyte–macrophage transformation and pathogeneses in native kidney disease in humans, with brief references to the major animal model studies. Rather than being exhaustive, we provide a general mechanistic framework, and against this background, emphasize recent findings, findings that remain controversial but pertinent to novel mechanistic paradigms, and those that may lead to therapeutic or prognostic insights applicable in renal diseases in humans.

## 2. Monocytes/Macrophages in Renal Pathology

Renal structure can be compartmentalized anatomically into glomerular, tubulointerstitial, and vascular. “Medical” renal diseases, i.e., those unrelated to neoplasms, tend to selectively involve individual compartments and are predominantly inflammatory (e.g., glomerulonephritis, tubulointerstitial nephritis, vasculitis). The inflammatory response in kidney disease is tightly regulated; inflammation may spontaneously resolve or be persistent with ensuing parenchymal scarring, depending on the underlying disease pathogenesis. Monocyte–macrophage lineage plays a critical role in the regulation of this inflammatory response in the kidney.

Although kidney diseases are of vastly different etiology and pathogeneses, a common feature is the infiltration of the affected renal tissue compartment by inflammatory cells, among which macrophages—detected by their markers CD68, CD14, and CD163—are a significant component. Macrophages are identified in most human renal diseases, suggesting a fundamental pathogenetic role across specific disease entities and affected tissue (Table 1 [3,4] and Table 2 [4,5,6,7,8,9]). Macrophages are also observed in several renal diseases in animal models used for pathogenetic investigation [3,4,5,6,7,8,9]. Table 3 [10,11,12], provides the immuno-phenotypic markers used to identify major macrophage morphologic subtypes. These markers, however, are not subtype-specific, with a complex overlap between markers and cell types, highlighting the challenges in the interpretation of functional significance [2,10].

### 2.1. General Mechanisms of Monocytes and Macrophages in Renal Disease Pathogenesis

The broad trajectory of monocyte–macrophage transformation in renal disease pathogeneses is explained in Figure 1 [2,3]. This pathogenetic pathway highlights a marked phenotypic and functional plasticity, transitioning from one lineage to another in the kidney. These transitions are mediated by feedback cycles of the chemokines and cytokines abundant within an inflammatory milieu. Significant knowledge in this area has been obtained through single-cell RNA sequencing and live cell imaging in experimental animal models [2].

### 2.2. Monocytes/Macrophages in Homeostasis and Renal Disease

#### 2.2.1. Classical and Non-Classical Monocytes: Identification and Function in the Kidney

It is important to identify different types of monocytes in order to understand their different roles in the pathogenesis of glomerulonephritis and tissue damage. Two broad subgroups of monocytes are recognized in the pathogenesis of renal diseases: classical and non-classical monocytes. Non-classical monocytes remain within the circulation, move continuously throughout the viscera, and help patrol the intravascular compartment; thus, they are also called “patrolling” monocytes (*PMos*). Classical monocytes {“(migratory monocyte)”} are programmed to exit the circulation to take up residence in the extravascular interstitial and parenchymal tissue, where several functions, including phagocytosis, are accomplished.

CD68 is the commonly used marker for the monocyte–macrophage lineage in tissue sections and flow cytometry, in both human and animal models [11]. CD68, however, does not distinguish between subtypes of monocytes or macrophages. For subtyping of classical and non-classical monocytes, CD16 and CD14 are commonly used in human studies, with their counterparts Ly6C and CX3CR1 in mouse models [11,13]. Classical monocytes are CD14^hi^CD16^neg^CX3CR1^lo^ or Ly6C^hi^; non-classical monocytes are CD14^lo^CD16^pos^ or CX3CR1^hi^. CD43 is another commonly used marker for non-classical monocytes [14,15]. These antibody markers have helped highlight the transition from non-classical to classical subtypes, with the identification of an intermediate form of monocytes during transition [16].

Significant progress has been made in understanding the molecular signals that transform the phagocytic classical monocytes into *PMos*. The chemokine axis acting through CCR2 in circulating cells and the tumor-necrosis factor pathway (TNF-TNFR2) in parenchymal cells enter into a CCR2-based autocrine feed-forward loop. This autocrine positive feedback loop amplifies renal inflammation. This regulates the homeostasis of monocyte subtypes [14]. Classical monocytes, however, undergo differentiation with marked plasticity. They have the ability to migrate through tissue and antigen-transport to hemato-lymphoid tissues (e.g., spleen and bone marrow) without further differentiation. Classical monocytes can also reversibly transform into non-classical *PMos*. They can maintain their phenotypic status during inflammation, while simultaneously differentiating into pro-inflammatory effector cells or a reparative phenotype based on the inflammatory milieu in the microenvironment.

#### 2.2.2. Macrophage Subtypes: Identification and Function in the Kidney

M1 and M2 macrophages refer to the pro- and anti-inflammatory phenotypic macrophage subtypes, respectively (Figure 2). Functionally M1 macrophages results in recruitment and activation of inflammatory cells and response while M2 macrophages has a role in regulation of tissue healing and resultant fibrosis [2,3,4]. M2 macrophages are further subclassified into M2a, M2b, and M2c [2,3]. IL-4 and IL-13 activate M2a macrophages; immune complexes, lipo-polysaccharides, and Fc-receptor ligands activate M2b; and IL-10, TGF-β, and glucocorticoids activate M2c. M2a subtype results in anti-inflammatory activity of T-helper type 2 lymphocytes regulating tissue healing and fibrosis, while M2b subtype activates T-helper type 2 lymphocytes and plays a role in immunomodulation. M2c macrophages results in immunosuppression [2,3].

The subtyping of M1 and M2 macrophages is derived from in vitro cell culture studies. These subtypes are in a state of constant flux in an in vivo environment. Based on in vitro and animal experimental models, immune markers for macrophage subtypes have been standardized for human studies. These are used for diagnostic and research studies in circulating inflammatory cells through flow cytometry or in vivo cellular/tissue localization in kidney biopsies with immunohistochemistry.

Immunohistochemical staining with CD68 is non-specific for macrophages and can show cross reactivity with stromal fibroblasts. CD16, CD32, CD80, and CD86 identify pro-inflammatory M1s, while CD163 and/or CD206 identify the anti-inflammatory M2 subtype in kidney biopsies. CD163-positive M2 macrophages assume a spindle-shaped morphology and cannot be recognized by light microscopy in the interstitium. Similarly, M1 macrophages are uncommon in healthy human kidney tissue and represent extravasated monocytes transforming into macrophages. In addition, common M1 markers CD80, CD81 and CD86 do not work in paraffin sections with IHC [4]. M2 macrophages have been shown to cause irreversible injury to terminally differentiated podocytes [17]. Notably, there is a dynamic conversion of M1 and M2 phenotypes during renal repair. An imbalance in this homeostasis, leading to increased M2 macrophages, is associated with an increase in fibrosis and poor renal outcome.

The increase in CD68-positive macrophage densities in both the cortex and medulla were found to have strong positive correlation with cortical scarring in the tubuo-interstitial region in several native kidney diseases in human studies. The increase in CD68 IHC positivity in renal biopsies had no correlation with levels of acute phase reactants or white cell count [4]. CD163-positive M2 macrophage density by IHC at the time of initial biopsy correlated with CD68 density and correlated with the risk of end stage renal disease (ESRD) in the follow-up, as emphasized by a 4-fold higher risk of ESRD for patients with higher than median macrophage density [4].

### 2.3. Dendritic Cells: Identification and Function in the Kidney

The role of dendritic cells overlaps with monocytes and macrophages, particularly in the renal interstitial region. Dendritic cells are one of the earliest and most critical antigen-presenting cells in innate immunity. Functionally, dendritic cells overlap with the monocyte-macrophage lineage of inflammatory cells in the kidney in the role of immunosurveillance and phagocytosis (Table 4) [3]. Phenotypically, dendritic cells have finger-like dendrites, unlike macrophages. In healthy conditions, dendritic cells are present in several extracellular foci in the interstitial region surrounding the tubules and arteries. Once dendritic cells identify injurious stimuli in the tubulointerstitial region, they function in parallel with *PMos* in the intravascular region in glomeruli and vessels to activate the inflammatory pathway through several cytokines. Dendritic cells play a critical role in activating T-lymphocytes and in immune modulation, while macrophages regulate renal fibrosis and tissue remodeling due to inflammation [18].

Much of what we know about the receptors and cytokines that regulate the functions of dendritic cells and macrophages was obtained from murine models. In nephrotoxic mice models, renal dendritic cells induce CD4+ T-cell proliferation with concurrent secretion of interferon-γ and IL-10 [19,20]. Ly6Gneg renal dendritic cells secrete most of the pro-inflammatory TNF-α and IL-12 mediators. In nephrotoxic models, tubulointerstitial DCs release IL-1β through activation of NACHT, LRR, and PYD domains containing protein 3 (NLRP3) inflammasome caspase pathway [21]. Dendritic cell depletion is a major therapeutic target to alleviate renal scarring, with successful results in nephrotoxic experimental models [22].

Immuno-modulation due to auto-immune, cancer and infections, results in the recruitment of migratory classical monocytes, as well as phagocytic cells such as neutrophils and M1 macrophages. Depending on the nature and persistence of the injurious stimulus, T-helper lymphocytes, B cells, and various subtypes of M2 macrophages are activated. M2 macrophages also evolve from bone marrow-derived myeloid cells and have plasticity towards macrophage–myofibroblast transition (MMT) through activation of highly versatile cytokines, such as TGF-β, transitioning from the acute inflammatory phase of injury to progressive fibrosis of the interstitium and reduced regeneration in tubules, resulting in tubular atrophy [23]. Table 4 highlights the overlapping immuno-phenotype and functions between dendritic cells and macrophages [3].

Against the background of a comprehensive understanding of the pathogenetic roles of monocytes/macrophages, novel findings are detailed below for specific renal diseases in which the pathogenetic roles of monocytes/macrophages have been established and remain an area of intense research interest.

### 2.4. Glomerulonephritis (GN): Immune Complex Deposits and the Mononuclear Phagocytic System

Immune deposits by themselves do not explain the severity of renal injury, as well as its progression. The disease manifestations due to immunoglobulins produced by B cells are not entirely ameliorated by targeted therapies against B cells, particularly in auto-immune diseases such as systemic lupus nephritis (SLEN). Additionally, several genetic mutations identified within SLEN have protein products with actions outside of the adaptive immune system. Similarly, certain types of Toll-like receptors (TLRs), such as TLR-7 and TLR-9, have been strongly implicated in SLEN. Experimental deletion of both TLR-7 and TLR-9 simultaneously in mouse models results in significant improvement in symptoms with a reduction of immune complex deposits in the kidney, as well as serum reduction in titers of anti-dsDNA and anti-nuclear antibodies [24].

Similarly, different mouse models demonstrate an increase in both circulating and intraglomerular *PMos* in the active phases of SLEN, while simultaneously showing the absence of any significant effect due to T and B cells [15]. Mouse studies on SLEN show that physiologic macrophages are essentially confined to the interstitium-surrounding arteries, while *PMos* are primarily in the intravascular space in the glomerulus [16]. Several experimental models have demonstrated increased monocytes in immune-complex-mediated glomerulonephritis before human studies [17]. Intravascular monocytes with unique phenotypes have also been reported in experimental models of crescentic immune-complex-mediated glomerulonephritis [15,16,17].

#### 2.4.1. Crescentic Glomerulonephritis

Monocytes and macrophages reportedly have a significant role in extra-capillary proliferation in crescentic glomerulonephritis, which manifests with rapidly progressive renal failure requiring an early diagnosis and prompt immunosuppression to salvage renal function (Figure 3). Notably, monocytes and macrophages have been correlated with greater severity of renal dysfunction and proteinuria in crescentic glomerulonephritis; simultaneous recovery has been correlated with the amelioration of these cell lineages [13]. Significant scientific evidence suggests that the monocyte–macrophage lineage plays a pathogenic role in the active phases of crescentic glomerulonephritis. Tremendous attention has been aimed at developing new therapeutic agents to reduce acute injury by the monocyte–macrophage lineage and thus prevent progression to chronicity [13,25].

#### 2.4.2. Classical and Non-Classical Monocytes in Immune-Complex-Mediated Glomerulonephritis

Classical monocytes (“migratory” monocytes) and macrophages are shown in high numbers in immune-mediated crescentic GN, while non-classical *PMos* are confined to the intravascular compartment [13,14,15,16,25]. *PMos* are characterized by their ability to move independently from the direction of blood flow, as well as an inherent ability to adhere to and crawl on the endothelium without transmigration through the basement membrane barrier of the capillaries. When *PMos* identify endothelial injury, such as in immune complex deposits in glomeruli, they adhere to the endothelium through CD11a/CD18 (β-integrins, LFA1 family). There is a chemokine axis through tumor-necrosis factor α (TNFα), interleukin 1β (IL-1β), CCL2, ICAM, and others that help retain these *PMos* in the glomerular foci with immune deposits. In the presence of an appropriate chemokine milieu from immune complexes, *PMos* are retained for a longer time and further stimulate chemotaxis of neutrophils and classical monocytes. *PMos* are thus among the earliest inflammatory cells to be identified in the focus of glomerulonephritis and are detected as early as day 2 in experimental settings.

The duration of *PMos* in the glomerulus, rather than their raw number, is associated with the severity of renal dysfunction. *PMos* numbers are not directly associated with the structural damage or integrity of the glomerular filtration barrier. However, their prolonged presence in the positive feedback milieu of TNFα and IL1β results in the recruitment and activation of phagocytic cells, such as neutrophils and classical monocyte–macrophage lineages, and thus in the corresponding amount of damage to the structural integrity of glomerulus.

#### 2.4.3. Classical and Non-Classical Monocytes in Pauci-Immune Crescentic Glomerulonephritis

Pauci-immune crescentic GN is typically seen in the setting of anti-neutrophil cytoplasmic antibody (ANCA)-mediated necrotizing glomerulonephritis with crescents. Experimental mouse models demonstrated the role of monocyte-macrophage subtypes in this setting. The experimental models utilized granulocyte-monocyte colony-stimulating factor (GM-CSF) to stimulate IL1β to activate classical monocytes highlighting the similarities between the pathways [13]. The findings were akin to those in immune-complex-mediated GN, with classical monocytes associated with necrotizing structural damage in the glomerulus.

### 2.5. Macrophages in Human Proliferative Glomerulonephritis

Macrophages stained with CD68 have helped in understanding the correlation between the density of macrophage infiltrate and progressive kidney dysfunction and chronicity. Assessment of glomerular cellularity in mesangial and endocapillary regions using CD68 immunohistochemistry (IHC) has been studied in proliferative glomerulonephritis, IgA nephropathy (Figure 4), and other immune-complex-mediated glomerulonephritis, such as infection-related glomerulonephritis and SLE nephritis (Table 1 and Table 2). There is sufficient evidence that an increase in CD68-positive macrophage infiltration is associated with the severity of renal damage in the active phase of proliferative GN and is also associated with poorer long-term renal outcomes [25]. We found evidence of CD68-positive macrophage infiltration in the foci of endocapillary hypercellularity in renal biopsies of patients with pauci-immune crescentic glomerulonephritis (Figure 5).

The role of macrophages and their activation of advanced glycation end products, reactive oxygen species, and inducible nitric oxide synthase (*iNOS*) have been studied in several glomerular diseases such as IgA nephropathy, diabetic nephropathy, and C3 glomerulonephritis [3,4,5,6,7,8,9,26].

### 2.6. Monocytes and Macrophages in Tubulointerstitial Injury

Proximal convoluted tubules (PCTs) of the kidney have high metabolic activity and play a critical role in reabsorption of the high volume of glomerular ultrafiltrate in the nephron. PCTs are vulnerable to repeated injury due to ischemic and toxic injury and necrosis. PCTs also have high regenerative capability and repair.

Acute tubular necrosis (ATN) refers to necrosis of the PCTs, manifesting as acute kidney injury, with a resultant decrease in urine output. ATN is one of the common causes of acute kidney injury (AKI), with particularly high morbidity and mortality in patients with other severe co-morbidities and in elderly patients with chronic diseases affecting the renal reserve. Ischemic and toxic injuries are common, and proximal convoluted tubules are extremely vulnerable because of their high metabolic activity, oxygen demand, ATP generation, and rapid reabsorption. Despite being a very common condition, studying ATN presents significant challenges. One challenge is that renal biopsies are seldom performed for a pure suspected cause of ATN but are always indicated for other concomitant primary causes. A second challenge is in the unequivocal identification and quantification of ATN in renal biopsies.

The renal tubular compartment is a close functional continuum of the interstitial region. Interstitial inflammation is a common pathology in most human diseases. Distal convoluted tubules and collecting ducts of the nephron are more prone to injury from renal casts and inflammation. The inflammatory pathway, as a cause of ATN, is studied extensively in animal models but is seldom recognized or considered in human renal biopsies. Thus, there is limited literature on inflammation-mediated ATN in humans. A study from Seoul, South Korea, reported a strong association of higher density M1 macrophages with higher stages of AKI and ATN, but most of these patients had recovery of renal function. Higher density of M2 macrophages at biopsy was strongly associated with a decline of glomerular filtration rate and renal failure on follow-up, though there was no correlation of M2 density with ATN or AKI stages [27]. Similarly, in a study on murine models from Cambridge University, the role of circulating B-cells secreting monocyte recruiting chemokine, such as CCl2, was reported in ATN. B cells produce CCl7 to facilitate chemotaxis of neutrophils and monocytes; blocking CCl7 reduced the severity of AKI, as well as the margination of inflammatory cells. Urine levels of CCl7 were also elevated in patients with AKI, highlighting the inflammatory pathway in ATN [28].

M2 macrophages are known to play a major role in the coordination of tubular regeneration, as well as in maintaining the integrity of renal tubules after injury, highlighting their critical role in ATN and AKI (Table 5) [3]. Several studies indicate that M1 macrophages are early responders in glomerulonephritis. With persistence of the injurious stimulus, M1 macrophages get transformed into different M2 subtypes. In addition, M2 macrophages originating from bone-marrow-derived myeloid cells also play a significant role in the macrophage to myofibroblast transformation, considered critical in interstitial fibrosis. We have observed the distribution of lymphocytes and macrophages in several patients with drug-induced interstitial nephritis (Figure 6). M2 macrophages interact with T-helper lineage lymphocytes in modulating the immune response in glomerular and tubulointerstitial regions. This dynamic interplay of different types of macrophages and monocytes is regulated by a series of inflammatory chemokines and cytokines, such as IL-6, TNF-α, TGF-β, CCL-2, and CCL-7, and a plethora of receptor pathways [3].

In addition to their prominent role in glomerular diseases, macrophages, especially M2 macrophages, have a dynamic regulatory role in the reparative response of tubular cells in conjunction with interstitial remodeling [17].

### 2.7. Monocytes and Macrophages in the Pathology of Renal (and Extra-Renal) Vasculature

Monocytes and macrophages are implicated in primary arterial hypertension and are a major area of research in targeted therapeutics. Kidneys are a major regulator of blood pressure due to the regulation of renin-angiotensin release, intravascular volume, and electrolytes regulation. Lys6C-positive monocytes and macrophages in the kidneys promote sodium reabsorption through the NKCC2 co-transporter. Experimental studies have shown that depletion of monocytes or GM-CSF-derived macrophages can prevent hypertension induced by angiotensin II, as well as vascular remodeling [29]. IL-6, an inflammatory cytokine produced by monocytes, has a strong association with arterial hypertension. IL-1 blockade has been shown in experimental studies to overcome this pathway of hypertension.

The role of monocytes and macrophages in the pathogenesis of atherosclerosis and remodeling of renal vasculature is well known [30]. This leads to increased arterial resistance and hypertension. In advanced atherosclerotic lesions, there is continuous recruitment of monocytes to the plaque, which may contribute to plaque rupture and instability [31].

The pro-inflammatory cytokines secreted by resident macrophages in the setting of chronic kidney disease are known to induce calcium crystallization and are a major causative factor implicated in the arterial calcification associated with cardiovascular diseases in the setting of chronic kidney disease [29,30]. Intimal calcification, which is a major determinant of plaque rupture, acute myocardial infarction, and medial calcification determines vascular stiffness, intra-arterial pressure (Monckeberg’s medial sclerosis), and valvular calcification. All of these have been associated with cytokine stimulus from the monocyte–macrophage lineage [31]. This pathway is considered one of the most important pathogeneses behind the high cardiovascular mortality in the setting of chronic kidney disease [29,30,31,32].

## 3. Targeting Macrophages as a Potential Therapeutic Opportunity in Glomerulonephritis

Animal models with AKI caused by ischemia-reperfusion injury, urine outflow obstruction, or crescentic and/or proliferative glomerulonephritis have provided valuable information on pathogenesis and potential therapeutic targets. Several major challenges have already been overcome in designing therapeutic targets. Initial challenges were in the identification of individual subtypes of inflammatory cells from the renal tissue. This constraint was easily overcome for circulating monocytes with multi-color flow cytometry. Identification of monocyte and macrophage subtypes have also improved with the advent of highly specific immunohistochemical markers. Together live cell imaging and single-cell RNA sequencing, led to a recognition of considerable plasticity among the different lineages in response to injurious stimuli. It was subsequently realized that immuno-phenotypic identification of monocyte–macrophage subsets is of limited value unless their functional characteristics are simultaneously studied. With further genetic, ultrastructural, and immunological studies, the exact cytokine mediators that regulate this plasticity and pathogenic tissue change were identified and several attempts have been made to selectively block these cytokines in experimental animal models to attenuate renal injury.

The other major challenges in this effort are the marked overlap of cytokines in their regulatory/inhibitory effect on various monocyte/macrophage lineage subtypes and the promiscuity of the various cytokines and their receptors in different cells and viscera. For instance, several attempts to target tyrosine kinase receptors in AKI were not successful due to high toxicity profiles. In view of this inadvertent toxicity, efforts have been redirected towards highly specific pathways regulating plasticity and activation of monocyte and macrophage subtypes, rather than targeting individual cytokines (Table 6) [16]. Significant strides have been made in regulating monocyte-macrophage homeostasis in therapeutic applications such as monocyte recruitment, monocyte proliferation and M1 to M2 subtype transition. There is extensive literature for each of these targets and the aim of this review is to highlight the potential translational research in each of these pathways. [33,34,35,36,37,38,39,40,41,42,43,44,45,46,47,48,49,50,51,52,53,54,55]. It is beyond the scope of this broad review to individually highlight each of these targets in detail.

A few studies from our own group exemplify the utility of this strategy. We have successfully demonstrated the beneficial effect of blocking folate receptor-β activity in macrophages in mouse models of experimentally induced anti-glomerular basement membrane (anti-GBM) disease. Though approved drugs such as methotrexate and its precursor aminopterin are already available to target folate receptor activity, they have significant toxicity and are not the mainstay of treatment. Hence, a novel folic acid-aminopterin conjugate (EC-2319) was designed, as a part of this study to selectively deliver the aminopterin molecule intracellularly through the folic acid receptor, thus overcoming the adverse effects. The compound was effective in ameliorating AKI and fibrosis in our experimental model [56].

We also targeted a novel independent pathway in the anti-inflammatory activity of A2A adenosine receptors in macrophages in experimentally induced anti-GBM disease. We successfully demonstrated the potential anti-inflammatory property of A2A adenosine receptors in reducing AKI, as well as progressive renal scarring in proliferative and crescentic glomerulonephritis. We used knock-out mouse models in comparison with wild-type animal models as controls [57].

## 4. Translational Value of Monocyte–Macrophage Axis in Diagnostic Clinical Nephro-Pathology

In a recent multicentric study of 324 renal biopsies from patients with a wide spectrum of 17 common renal conditions, it was found that an increase in the median count of CD68-positive macrophages and CD-163-positive M2 macrophages in the renal cortex and medulla was associated with reduced renal function both at the time of biopsy as well as poor renal outcome at the long-term follow-up [4]. The increase in cortical CD68-positive macrophages was associated with increased interstitial fibrosis (IF) and tubular atrophy (TA) in renal cortex.

In patients with systemic lupus erythematosus and small vessel vasculitis, the densities of CD68-positive macrophages and CD-163-positive M2 macrophages were associated with disease activity [4]. Similar results were observed with the higher density of CD68- and CD163-positive macrophages in the medulla in cases of upper urinary tract infections. CD14-positive monocytes had no correlation with renal function impairment at the biopsy.

In a study from Peking University hospital, urinary soluble CD163 (UsCD163) levels were found to be elevated in patients with diffuse proliferative glomerulonephritis and also with cellular crescent formation. There was no correlation of UsCD163 levels with acute tubular necrosis and tubulo-interstitial nephritis. This feature highlights the potential diagnostic utility of M2 macrophages in renal diseases [58].

Endocapillary hypercellularity is a major histopathologic feature in the activity classification of class III and class IV lupus nephritis. However, this feature has high inter-observer reproducibility [59]. Endocapillary hypercellularity is characterized by luminal occlusion of glomerular capillaries due to infiltration by inflammatory cells. Monocytes have been reported in significant numbers in immune complex mediated glomerulonephritis such as lupus nephritis and IgA nephropathy [60,61]. CD68-positive macrophages were utilized as a surrogate for endocapillary hypercellularity of lupus nephritis in a multi-centric study performed in the Netherlands and United Kingdom. In an interesting observation, they found that glomeruli with ≥7 CD-68 positive macrophages had a good concordance for endocapillary hypercellularity in lupus nephritis. The modified activity index utilizing CD-68 positive macrophages was found to have significant correlation with the estimated glomerular filtration rate (eGFR), as well as with the current activity index assessment by histological assessment [7,62]. 

The importance of CD-68-positive macrophages as a useful surrogate marker for endocapillary hypercellularity was earlier studied from Oxford University, in IgA nephropathy (IgAN) [6]. This study was conducted with the backdrop of Oxford classification for prognostic evaluation of IgAN, based on the evaluation of histological parameters of mesangial and endocapillary hypercellularity, segmental sclerosis and tubular atrophy/interstitial fibrosis (MEST score). The Oxford scoring system for classification of IgAN and its prognostic utility was originally proposed by the working group of International IgAN and Renal Pathology Society, in the year 2009 [63]. This scoring system has been extensively validated in several countries from across the world [64,65,66,67,68,69,70,71,72,73]. Based on scientific literature from all over the world, the Oxford scoring system incorporated crescents into the MEST score, transforming it into a composite C-MEST score [74].

In the validation studies from across the globe, endocapillary hypercellularity had a higher interobserver variability [60,61]. In this context, glomeruli with >6 CD-68 was found to be a useful threshold for identification of endocapillary hypercellularity. Immunohistochemical staining with CD68 also enabled digital quantification of the macrophages with image analysis and the correlation was found to be robust. In the same study, the authors found moderate correlation of tubulointerstitial CD-68-positive macrophages with tubular atrophy and interstitial fibrosis (r = 0.59, r^2^ = 0.35, *p* < 0.001) [7].

In a study from Germany, macrophage subtypes were studied using immunohistochemistry in renal biopsies of patients with lupus nephritis. CD-68-positive macrophages were further categorized as M1 (iNOS/CD68), M2a (CD206/68), and M2c (CD163/CD68) subtypes. They found that CD68-positive-macrophages were higher in the patients with active class IV lupus nephritis in comparison with class II and class V lupus nephritis. Higher density of macrophage infiltration of both M2a and M2c subtypes in the tubulointerstitial region had significant correlation with serum creatinine levels. A higher number of M2a and M2c macrophages in the tubulointerstitial region had significant correlation with hypertension, and interstitial fibrosis/tubular atrophy, respectively [75].

All this recent evidence highlights the increasing interest in validation of the monocyte–macrophage lineage in diagnostic as well as prognostic application in renal biopsies from patients with glomerular as well as tubulointerstitial diseases. CD68-positive macrophages can help in objective detection of endocapillary hypercellularity in proliferative glomerulonephritis, highlighting severe and active disease and its potential as a diagnostic biomarker. Increased density of both CD68-positive macrophages and the CD163-positive M2 macrophage subtype have been associated with progressive chronic renal injury and poorer outcomes, and can be a useful prognostic biomarker. On the contrary, monocyte detection using CD14 in renal biopsies has not been found to have diagnostic or prognostic utility. This will need further human validation studies with other markers.

## 5. Conclusions and Future Directions

The monocyte–macrophage axis plays a significant role in immune modulation and healing in a wide spectrum of renal diseases. Significant progress has been made in the diagnostic capability and understanding of its role in etiopathogenesis and common effector pathways in cellular and tissue healing and fibrosis. The emphasis on diagnostic utilization of elevated urine M1:M2 ratio in acute tubule-interstitial nephritis in comparison with decreased M1:M2 ratio in patients with crescentic glomerulonephritis was demonstrated in a recent human study, highlighting urinary detection of macrophage subtypes as liquid biopsy [58].

This review provides a comprehensive overview of the critical role of the monocyte-macrophage axis in human glomerular diseases and immune modulation. Immune complex deposits and inflammatory pathways are the most critical pathogenetic pathways in most glomerular diseases, and the importance of the monocyte–macrophage axis to this pathogenic pathway still needs extensive study. Most of the existing knowledge on human diseases was preceded by meticulous animal experimental models, highlighted in this review. We summarized the important studies demonstrating the role of the monocyte–macrophage axis in glomerular, tubulointerstitial and vascular injury, and its role in matrix remodeling and parenchymal scarring. New evidence points to the role of this axis both in immune complex, complement-mediated as well as pauci-immune glomerulonephritis. These advances provide us with promising avenues for targeted therapy towards enhanced immune response with minimal fibrosis, preserving the function of the nephron and the kidney. There are still significant knowledge gaps and scope for future research in human renal diseases focusing on the role of the monocyte–macrophage axis, as there is heterogeneity in treatment response and prognosis. Continued research enables us to refine our diagnostic, prognostic, and therapeutic strategies and lead focused targeted therapy for renal diseases.

## Figures and Tables

**Figure 1 ijms-25-03763-f001:**
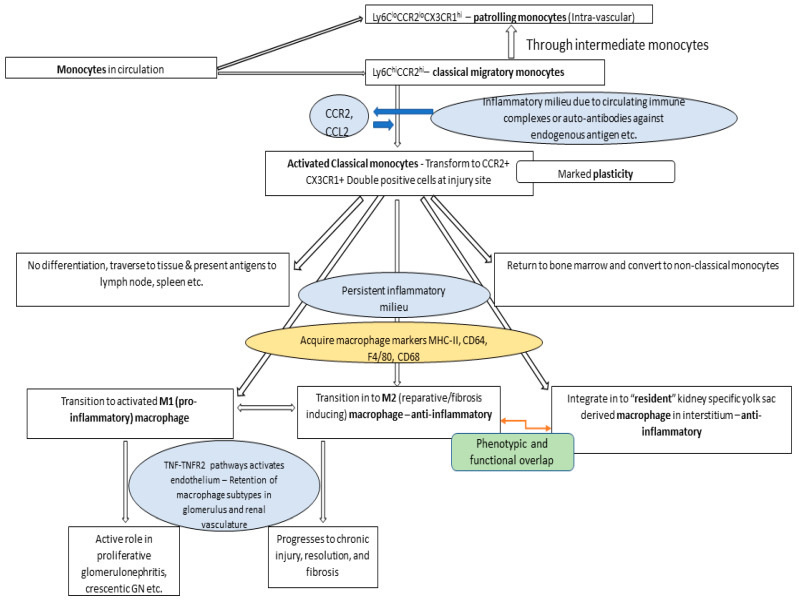
General mechanisms underlying the monocyte–macrophage signaling axis during renal inflammation [2,3]. Key cytokines regulating the differentiation and functional roles of the monocyte–macrophage lineage are shown. The immunomodulatory roles of the monocyte–macrophage lineage across various anatomical and functional compartments of the kidney are underscored by its intricate involvement in renal inflammation.

**Figure 2 ijms-25-03763-f002:**
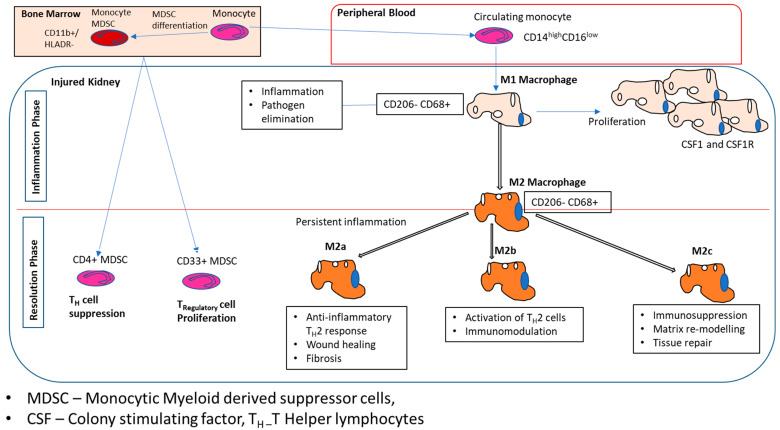
Function of marrow-derived monocytes and macrophages in kidney pathology. The regulatory mechanisms orchestrating the recruitment of marrow-derived monocytes and macrophages to the renal parenchyma during injury are shown [17].

**Figure 3 ijms-25-03763-f003:**
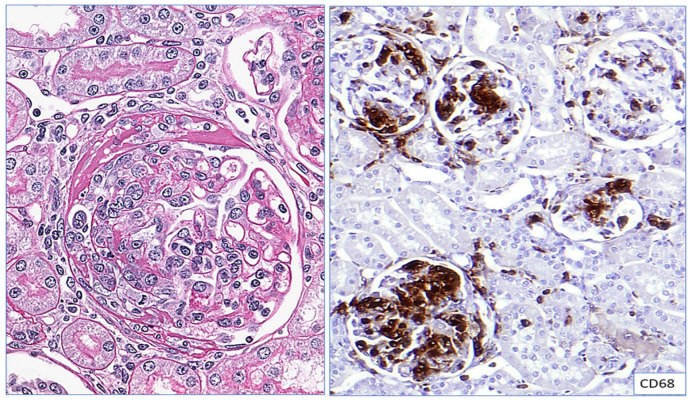
CD68 in crescentic experimental rat models: Section shows cellular crescent in a glomerulus from experimental rat models with prominent extra capillary proliferation in the Bowman’s space, partially obliterating the glomerular tuft (**left**—Hematoxylin and Eosin stain, ×400). Section on the right pane highlights CD68-positive macrophages in the glomeruli from the crescentic glomerulonephritis in rat model (**right**—diaminobenzidine stain, immunohistochemistry with CD68, ×200). Crescentic glomerulonephritis indicates a severe acute form of glomerulonephritis and the image highlights strong CD68-positive macrophage infiltration in both intra-glomerular as well as extra-capillary crescentic proliferation in our experimental rat model studies. Higher expression of CD68 in crescentic glomerulonephritis was correlated with rapid decline in renal function and higher degree of proteinuria.

**Figure 4 ijms-25-03763-f004:**
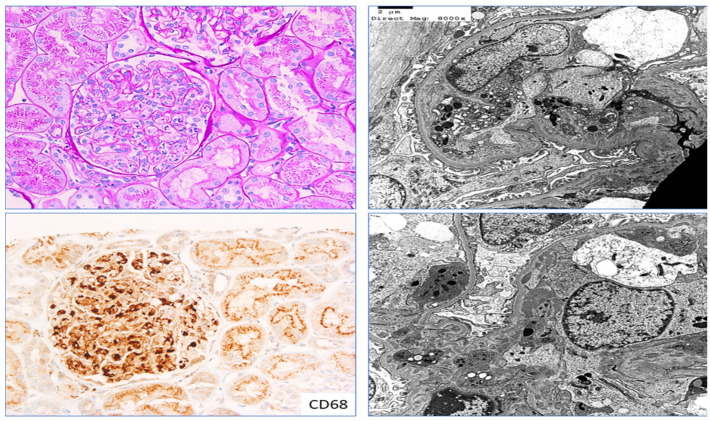
CD68 in IgA nephropathy: Section shows prominent mesangial and endocapillary hypercellularity in a glomerulus with proliferative glomerulonephritis in a patient with IgA nephropathy (**left top**; Periodic acid–Schiff stain, ×200); section in the **left bottom** pane highlights CD68-positive macrophages from the same glomerulus (indirect immunohistochemistry with CD68 antibody, diaminobenzidine stain); sections in the **right top** and **bottom** pane highlights ultrastructural details of macrophage infiltration in the endocapillary and mesangial region of the same glomerulus, respectively (transmission electron microscopy, uranyl acetate stain). In the example given above, IgA nephropathy is cited as an example of immune-complex-mediated glomerulonephritis where prominent macrophage infiltration is identified in the endocapillary and mesangial region. Increase in endocapillary and mesangial hypercellularity is associated with higher activity of the disease having greater renal dysfunction and is used in Oxford MEST-C classification prognostic scoring system of IgA nephropathy.

**Figure 5 ijms-25-03763-f005:**
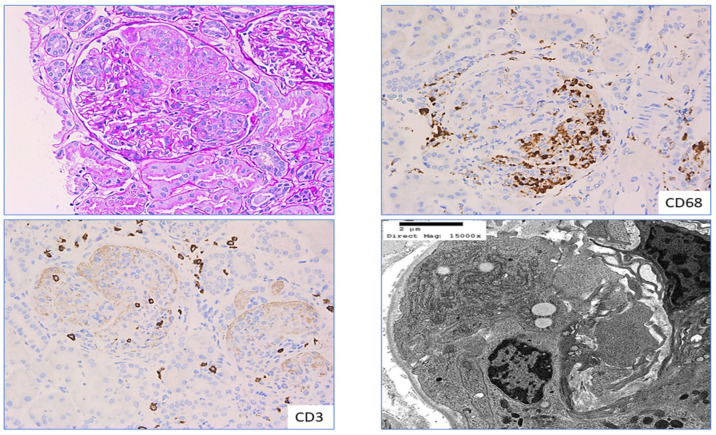
CD3 and CD68 in pauci-immune crescentic glomerulonephritis: Section from a patient with pauci-immune crescentic glomerulonephritis shows cellular crescent in a glomerulus with collapsed tuft exhibiting focal fibrinoid necrosis (**left top** pane); **left bottom** pane and **right top** pane highlights infiltration of CD3-positive T lymphocytes and CD68-positive macrophages in the same glomerulus, ×200; **right bottom** pane highlights ultrastructure of macrophage infiltration in the capillary loop of the glomerulus from the same patient.

**Figure 6 ijms-25-03763-f006:**
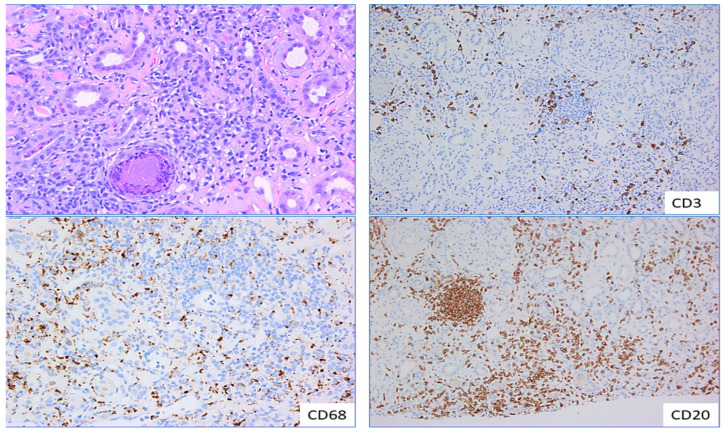
Monocytes and macrophages in tubulointerstitial nephritis.—Sections from a patient with drug-induced tubulo-interstitial nephritis highlights the density of lympho-plasmacytic and histiocytic infiltration in the renal interstitium, ×200 (**left top** pane). Images in the **left bottom** and **right** pane highlights immunoperoxidase stains with CD68, CD3 and CD20 demonstrating the distribution of macrophages, T and B lymphocytes, respectively in the interstitial infiltrate, ×100.

**Table 1 ijms-25-03763-t001:** Major renal diseases highlighting the role of CD-68-positive macrophages in human and animal studies with significant prognostic association [3,4].

Number	Renal Disease	Renal Compartment Semi-Quantitively Assessed for CD-68 Positive Macrophages
1	IgA nephropathy [4]	Endocapillary hypercellularity [4]
2	Systemic lupus erythematosus nephritis [4]	Endocapillary hypercellularity and tubulo-interstitial inflammation [4]
3	Crescentic glomerulonephritis * [3,4]	Extra-capillary macrophage infiltration [3,4]
4	Tubulo-interstitial toxicity * [3]	Tubulo-interstitial macrophage infiltration [3]

* CD-68 has been studied in experimental crescentic glomerulonephritis and tubulointerstitial nephrotoxic animal models as well as in few human studies.

**Table 2 ijms-25-03763-t002:** Renal diseases with CD-14-positive monocytes and CD-68- and CD-163-positive macrophages reported in human studies in correlation with renal outcome on follow-up [4,5,6,7,8,9].

Number	Name of the Disease Entity	CD-68-Positive Macrophage Density in Renal Cortex	CD-163-Positive Macrophage Density in Both Renal Cortex and Medulla	CD-14-Positive Monocyte Density in Renal Cortex
1	Small vessel vasculitis [4]	Higher density of CD-68 in cortex and CD-163-positive macrophages in both cortex and medulla compared to healthy controls. High density of cortical CD-68-positive macrophage predicts shorter renal survival in all diseases collectively as well as in each disease independently.		No difference between cases and controls No correlation with renal function
2	IgA Nephropathy [4,5,6]			
3	Hypertension [4]			
4	Membranous glomerulonephritis [4]			
5	Focal segmental glomerulosclerosis [4]			
6	Thrombotic micro-angiopathy [4]			
7	Minimal change disease [4]			
8	Tubulo-interstitial nephritis [4]			
9	Diabetic nephropathy [4]			
10	Systemic lupus erythematosus [4,7,8,9]			
11	Urinary tract infection [4]			
12	Amyloidosis [4]			
13	Post-infectious glomerulonephritis [4,9]			
14	Thin basement membrane disease—Alport disease [4]			
15	Other diseases * [4]			

* Other diseases [4]—C1q/Mesangio-proliferative GN, C3GN, para-infectious mesangio-proliferative GN, light chain nephropathy, nicotine induced nodular glomerulosclerosis, lupus like nephritis, feto-fetal-transfusion syndrome, associated multi-organ failure, atypical hemolytic uremic syndrome, tubular dysgenesis. Note: CD-68, CD-163 and CD-14 have been studied extensively in animal models and are now validated for diagnostic and prognostic biomarker application in human biopsies as well as for research. CD-68 and CD-163 have cytoplasmic expression while CD-14 has a nuclear expression. Recent evidence highlights the presence of CD-14 in clear cell renal cell carcinoma cells in addition to the monocyte lineage. Higher CD-14 expression in the peri-tumoral immune infiltrate is associated with poorer prognosis. Several studies have documented high density of CD-68-positive macrophages in tubulo-interstitial region in correlation with poor long term renal survival in IgA nephropathy and SLE nephritis [4,5,6,7,8,9].

**Table 3 ijms-25-03763-t003:** Markers of monocyte/macrophage lineage studied in kidney diseases [10,11,12].

Cell Lineage	Markers
Monocytes/macrophages from hematopoietic stem cells [10,11]	CD11b^+^, CSF1R^+^, Ly-6C^+^, Ly-6G^−^, CX3CR1^+^, CCR2^+^
Resident macrophages in kidney [10,11] (Erythro-myeloid progenitors from yolk sac)	CD45^+^, CD 11b(low), F4/80 (high), Ly-6C(low)
Dendritic cells—CD-103-positive subtype [10,11]	CD11c^+^, MHC-II^+^, CD 103^+^, CD11b(low), CX3CR1-, SIRPα-
Dendritic cells—CD-11b-positive subtype [10,11]	CD11b^+^, CD103^−^, CX3CR1^+^, SIRPα^+^,
Classical monocytes in peripheral blood (Ly6C^+^ MHCII^−^) [10,11]	CD43(low), CCR2^+^, CX3CR1(low), CD62L^+^, TREML4^+/−^, MHCII^+/−^
Non-classical patrolling monocytes (LY6C^−^ MHC II low/^−^) [10,11]	CD43(high), CCR2^−^, CX3CR1(high), CD62L^−^, TREML4^+^, MHC II^+/−^
M2a macrophages [12]	CD209
M2b macrophages [12]	CD86
M2c macrophages [12]	CD163

All the above-mentioned immune markers are initially standardized in animal models and subsequently validated in human research and diagnostic applications.

**Table 4 ijms-25-03763-t004:** Overlapping phenotype and functions of dendritic cells and macrophages. The structural and functional similarity between dendritic cells and macrophages is highlighted, with an emphasis on the shared and unique antigens, receptors, and cytokines that regulate their development and function. Expansion of abbreviation of transcription factors regulating macrophages and dendritic cells—IRF4 and IRF5—Interferon regulatory factors 4 and 5 (DNA binding transcription factor involved in inflammatory mediation), STAT3 (Signal transducer and activator of transcription 3), ID2 (Transcription regulator, activated by proinflammatory cytokines; a helix-loop-helix protein that inhibits the E protein transcription factors E2A, HEB and E2-2. The E proteins play important roles in B lymphocyte and T lymphocyte lineage specification and commitment, ATF3—master regulating activating transcription factor 3, involved in modulating metabolism, immunity and oncogenesis, ZBTB46—zinc finger and BTB containing domain 46 (transcription factor selectively expressed by classical dendritic cells) [3].

Macrophages	Dendritic Reticulum Cells
Functions—Tissue surveillance, secretes cytokines and chemokines, phagocytosis and cytotoxicity, fibrosis and matrix remodeling	Functions—Tissue surveillance, secretes cytokines and chemokines, phagocytosis, antigen presentation, T cell stimulation and immune tolerance
Transcription factors regulating macrophages—IRF4, STAT3 and IRF5	Transcription factors regulating dendritic cells—ID2, IRF8, ZBTB46, B-ATF-3
Surface markers of inflammatory M1 macrophages—Ly6G, Ly6C, CD62L	Surface markers of dendritic cells— CCR7, CD103, CX3CR1, CD135, CD1c, CD209
Surface markers of anti-inflammatory M2 macrophages—IL4R/IL10R, CD206, CD163, CD68, CSF-1R, CD14, CD16, CD54, CD32	
Shared antigens between macrophages and dendritic cells—CD80 (B7.1), SIRP α, F4/80 (EMR1), MHC II, CD11b, CD86	

**Table 5 ijms-25-03763-t005:** Type 1 and type 2 macrophages in the interstitium of kidney [3,12,18].

Macrophage Cell Types	Role in Renal Disease
M1 macrophages [3]	M1 macrophages are derived from monocytes which are recruited in renal interstitium due to the influence of adhesion and pro-inflammatory mediators ICAM1, osteopontin, etc. M1 macrophages are implicated in direct renal injury at the level of proximal and distal tubules, as well as in enhancing antigen presentation to T lymphocytes, mediated by IL-12 and IL-23
M1 to M2 macrophage transition in interstitium of kidney [3]	c. Facilitated by macrophage colony stimulating factor and IL-10.
M2 macrophages [3]	d. Repair of renal tubular injury mediated by Wnt7b and IL22 e. Resolution of tubular and interstitial inflammation mediated by IL-10 and HO-1 f. M2 macrophages proposed to play a role in trans-differentiation of pericytes, and myofibroblasts resulting in increased collagen matrix deposition and interstitial fibrosis. These changes are mediated by TGFβ, PDGFβ and galectin 3
M2a macrophages [12,18]	g. Activate M2b macrophages, predominantly through T helper cell mediated pathways
M2b macrophages [12,18]	h. Plays a role in immunoregulation
M2c macrophages [12,18]	i. Take part in matrix remodeling due to inflammation, tissue repair, phagocytosis of degenerated and dead cells

Abbreviations:ICAM1—intercellular adhesion molecule 1, IL—Interleukins, HO1—Haemoxygenase 1, Wnt—wingless related integration site, TGFβ—Transforming growth factor beta, and PDGFβ—platelet derived growth factor beta.

**Table 6 ijms-25-03763-t006:** Targeted compounds focused on monocyte–macrophage axis in renal diseases in various stages of clinical trials [17,33,34,35,36,37,38,39,40,41,42,43,44,45,46,47,48,49,50,51,52,53,54,55].

Level of Target in the Monocyte–Macrophage Activation in Renal Disease Pathogenesis	Compounds in Clinical Trials
Recruitment of circulating monocytes in blood to injured site in renal parenchyma	CCR2 antagonist (phase III in diabetic kidney disease) [33] CCL2, CCL5, CXCL16 and CCL21 inhibition [33,34,35,36] CCR2, CXCR6, CCR7 and CX3CR1 inhibition [35,36,37,38,39,40] C3aR and C5aR inhibition [41,42,43] SYK inhibition (phase II in IgA Nephropathy) [44]
Monocyte proliferation	CSF1R inhibitor [45]
Inhibition of monocyte to M1 macrophage transition and proliferation of M1 macrophages	JNK inhibitor [46] SYK inhibition (phase II in IgA Nephropathy [44,47]
M1 macrophage to M2 macrophage transition	JAK-STAT inhibitor (phase II in Diabetic kidney disease) [48,49] IL-4R and IL-13R [48,49] Galectin 3 inhibitor (phase IIa in Diabetic kidney disease) [50,51,52]
M2 macrophage to myofibroblast transition	Src inhibitor [53] JAK-STAT inhibitor [54] Smad3 inhibitor [55] Asiatic acid and naringenin [38]

Abbreviations used: CCR—chemokine receptor, CCL—chemokine ligands, CXCL—CXC family of chemokine ligands, C3R and C5R—complement C3 and C5 receptors, SYK—novel spleen tyrosine kinase inhibitors, CSF1R—colony stimulating factor 1 receptor, JNK inhibitor–C-Jun N-terminal kinase inhibitor, JAK-STAT—Janus kinase/signal transducers and activators of transcription, IL—interleukins, SRC inhibitor—SRC family of tyrosine protein kinase inhibitor, Smad3—a family of smad proteins derived from the fusion of Caenorhabditis elegans Sma genes and the Drosophila Mad (mothers against decapentaplegic) proteins to transduce signals.

## Data Availability

Data is available on request from the authors.

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
