# Peer review of "Monocytes and Macrophages in Kidney Disease and Homeostasis"

_ijms, 2024, doi:10.3390/ijms25073763_

Round 1

Reviewer 1 Report (Previous Reviewer 2)

Comments and Suggestions for Authors

The authors have done very little to revise this manuscript from the original version submitted in December 2023. Review articles are meant to include a comprehensive set of past publications on a topic and as such may have over 100 references. This version of the manuscript has few new references added. When references were requested to be included WITHIN tables, they were added in a group in a note beneath the table. 

Author Response

Reviewer 2 Report (New Reviewer)

Comments and Suggestions for Authors

The manuscript reviewed macrophage-induced pathogenetic mechanism as a background for the latest findings about macrophages’ roles in different renal compartments and common renal diseases. It is well-written and very detailed. The manuscript contains some errors and seems that it is not its final version, therefore it is difficult to make a final decision and I am not sure if the manuscript is finished. My comments are listed below:

·         Why is most of the text highlighted and some of the references marked red?

·         In my opinion, the section list is unnecessary in such a short manuscript.

·         The data given in lines 85-92 are not supported by the references. As the manuscript is classified as a review, citations must be placed after the. 

·         Line 119: lack of number next to “[Ref:”

·         Is the Figure 1 cited from the literature or created by the authors? There are references [2,3] cited in the text, whereas the figure caption does not include any references.

·         What is the final title of the manuscript? The title in the manuscript (“Monocytes and Macrophages in Kidney Disease and Homeostasis”) is different than the title in the reviewer’s system (“Human and Animal Monocytes and Macrophages in Homeostasis and Disease of the Kidney”).

Author Response

Response to reviewer 2

Reviewer comment

Author response

The manuscript reviewed macrophage-induced pathogenetic mechanism as a background for the latest findings about macrophages’ roles in different renal compartments and common renal diseases. It is well-written and very detailed. The manuscript contains some errors and seems that it is not its final version, therefore it is difficult to make a final decision and I am not sure if the manuscript is finished. My comments are listed below:

We thank the reviewer for a detailed and meticulous review.

We apologize for the errors and have corrected all those which have been pointed out.

·     Why is most of the text highlighted and some of the references marked red?

The texts were highlighted as a response to previous review process and the same is true for the references.

We have removed all the highlighting text to make it uniform, except for the new corrections.

·     In my opinion, the section list is unnecessary in such a short manuscript.

Authors are willing to remove the selection list, if approved by the editorial team. We have numbered it as selections as part of the journal recommendation.

      The data given in lines 85-92 are not supported by the references. As the manuscript is classified as a review, citations must be placed after the. 

The data given in lines 85-92 are cited as suggested

            Line 119: lack of number next to “[Ref:”

References have been added.

             Is the Figure 1 cited from the literature or created by the authors? There are references [2,3] cited in the text, whereas the figure caption does not include any references.

Figure 1 is created by the authors, to provide a broad overview of the review article and it includes information from most of the references cited in the article. Major reference for this figure is based on references 2 and 3 and the same is cited as suggested.

           What is the final title of the manuscript? The title in the manuscript (“Monocytes and Macrophages in Kidney Disease and Homeostasis”) is different than the title in the reviewer’s system (“Human and Animal Monocytes and Macrophages in Homeostasis and Disease of the Kidney”).

The final title of the manuscript is “Monocytes and Macrophages in Kidney Disease and Homeostasis”

The earlier title “Human and Animal Monocytes and Macrophages in Homeostasis and Disease of the Kidney” was modified as some of the earlier reviewers objected to mention of Human and animal in the title as confusing. This manuscript is primarily focused on the human studies on the role of monocytes-macrophage lineage in renal diseases, while highlighting the major animal model studies behind the understanding of the etio-pathogenesis. This is because all the authors in the review are senior professors working in the field of diagnostic renal pathology in hospital settings for humans, with research interest in animal models to understand the patho-physiology of several renal diseases.

Reviewer 3 Report (New Reviewer)

Comments and Suggestions for Authors

The review offers a clear and reasonably comprehensive overview. But we believe that the title would be more appropriate if changed in: “Monocyte and macrophage in glomerulonephritis and homeostasis”. In fact, in the review there are only indirect hints on the role of monocyte and macrophage in kidney transplantation. This topic still far from being deeply understood, can be considered very relevant since their potential application as therapeutics to prevent or treat allograft rejection, as well as challenges in their clinical translation.

Despite this, the discussion on the diverse roles of monocytes and macrophages including their phenotypic variability and their pathogenic roles, adds valuable insights to our understanding of renal pathophysiology.

Author Response

Authors sincerely thank the reviewer for the encouraging comments.

There is limited body of literature on this topic and still a lot of work needs to be done, particularly in human diseases.

We have tried our best to highlight some of the newer literature on human diseases and correlated them with the pathogenetic insight from animal model studies and have tried to explain it as succinctly as possible, based on our own experience

Reviewer 4 Report (New Reviewer)

Comments and Suggestions for Authors

This review articles described Monocytes and Macrophages in Kidney Disease and Homeostasis. It would be a good introduction for people engaging in the clinical and research fields related with kidney disease. However, this reviewer feels that the quality of manuscripts needs to be improved. It is not easy to track the definition of several cells, as it is changed in sections. Texts and figures showing morphological image do not match. Please also check whether the definition of macrophage subsets reflects the latest publications. It is important to include more recent publications to advance knowledge related with the title.   

(1) Table 1A: Please clarify this table shows information about human only, or human and animals.

(2) CD11b+ :  + should be superscript. Many other CD markers are also indicated without superscript.

(3) Table 2: Please integrate format of marker. CD45 + is missing. CX3CR1low; a space is not necessary between CX3CR1 and low? What is +/-?

(4) Line 142: Please define “patrolling” monocytes. The authors described that non-classical monocytes are also called “patrolling” monocytes (PMos). But in Line 157, it is described “the molecular signals that transform the phagocytic classical monocytes in to PMos. It means that non-classical monocytes and PMos is different cell populations.

(5) Line 152: CD14hiCD16negCX3CR1lo or Ly6Chi; Ly6Ch means cells of Ly6C single positive in CD63 positive population ?

(6) Line 158 to 159: Please rephrase the texts to be more readable.

(7) Fig.2: In recent review articles, M2a mediates Th2 inflammation. Please comment about it.

(8) Fig.3: Please check if the function of tissue surveillance is correct. What is dendritic reticulum cells?

(9) Fig. 3: Second column: IRAF4, Stat3 and other molecules are not chemokines.

(10) Fig. 3: CD209 is highly expressed in macrophages. F4/80 is not MHC class II. Please check other indications to see all information is correct. The reference is from 2014 and is old.

(11) Fig. 4: Legend title should be added. If not, the reference should be indicated.

(12) Line 272-274: The authors appear to tend explaining Fig. 4. But the image and texts do not match.

(13) Line 289: There is another definition of Classical monocytes (“migratory” monocytes). It is confusable. It is better to define cell populations in the first section to read this article more easily.

(14) Line 290: What is non-classical PMos?

(15) Fig. 5: Legend title should be added. This figure was prepared by the authors? If not, the reference should be indicated. It is the same to other majority of figures showing images of histology.

(16) Line 303-309: The authors appear to tend explaining Fig. 5. But the image and texts do not match.

(17) Line 324: Fig. 6 does not show the presence of monocytes.

(18) Table 3: In Fig. 2, the authors described role of M2 subsets. It would be better to describe M2 subsets, not just M2, in this Table, if possible.

(19) Table 4: The reference for each therapeutic strategy should be included.

(20) Lines 485 to 489: There is similar description in the first chapter. It is redundant and should be rephrased.

(21)This section of Translational value of Monocyte-macrophage axis in diagnostic clinical nephropathology just explains data of other research groups. It is important to propose how the previous finding leads to novel diagnostic strategy.

Round 2

Reviewer 4 Report (New Reviewer)

Comments and Suggestions for Authors

The revision improved the quality of manuscript. The authors should additionally check format of manuscript carefully and whether the latest publications are properly cited. Followings are just examples. 

(1) Table 3. The location of "a" to "i" should be adjusted in a line.

(2) Table 2. Dots should be adjusted in a line.

(3) Between line 495 and 496: space is needed.

(4) CD-68 in Line 539:hyphen is required?

(5) The authors added around 25 references. However, all are old manuscripts published before 2018 and did not include the latest publications in the last five years. Is it appropriate?

Author Response

Reviewer comment

Author response

The revision improved the quality of manuscript. The authors should additionally check format of manuscript carefully and whether the latest publications are properly cited. Followings are just examples. 

Authors sincerely thank the reviewer for detailed assessment. We have carefully checked all the references cited and the format of the manuscript.

 Table 3. The location of "a" to "i" should be adjusted in a line

The corrections have been made in the manuscript

Table 2. Dots should be adjusted in a line.

The corrections have been made in the manuscript

Between line 495 and 496: space is needed.

Space has been added in the manuscript

CD-68 in Line 539: hyphen is required?

Hyphen has been added for CD-68

The authors added around 25 references. However, all are old manuscripts published before 2018 and did not include the latest publications in the last five years. Is it appropriate?

Authors sincerely thank the reviewer. In-depth review of all the latest clinical trials and original articles have been done.

The clinical trials are still on-going and we did not have original research with published results having shared data.

Most recent human studies on the topic are cited in references 4,5,6,7,8,9, 27, 29, 30,31,32,33,34, 61,63,65, 67,68,69, 71 and 81.

The most recent important original basic research work on the field is cited in Ref. no. 12, 14,

          12. S Zhao, M Si, X Deng, D Wang, L Kong,  

           Q Zhang. HCV inhibits M2a, M2b and M2c

           macrophage polarization via HCV core

           protein engagement with Toll like

           receptor-2. Experimental and Therapeutic

           Medicine 2022;

           https://doi.org/10.3892/etm.2022.11448

14.   Turner-Stokes T, Diaz AG, Pinheiro D, Prendecki M, McAdoo SP, Roufosse C et al., Live imaging of monocyte subsets in immune complex mediated glomerulonephritis reveals distinct phenotypes and effector functions. Journal of American Society of Nephrology 2020; 31: 2523-2542.

Few recent review articles are found in the literature but we wanted to minimize adding too many review articles in our citations, as it will dilute the quality of our review without additional original research reference citations.

For example, a latest reference review article was not cited,

a.     Hofherr, A., Williams, J., Gan, LM. et al. Targeting inflammation for the treatment of Diabetic Kidney Disease: a five-compartment mechanistic model. BMC Nephrol 23, 208 (2022). https://doi.org/10.1186/s12882-022-02794-8

The most recent clinical trial for which the data has been submitted and is still under review by FDA is enlisted below,

a.       ClinicalTrials.gov Identifier: NCT02768948 - Telmisartan Promotes the Differentiation of Monocytes Into Macrophages M2 in Diabetic Nephropathy?

We have exhaustively studied the literature to ensure that the review provides updated and shares scientifically valid information and hope that our response will be considered.

This manuscript is a resubmission of an earlier submission. The following is a list of the peer review reports and author responses from that submission.

Round 1

Reviewer 1 Report

Comments and Suggestions for Authors

The authors have significantly increased the length of the Discussion and number of citations.  The extension to the Discussion includes reference to several relevant studies, although an overarching analysis of these references could be improved.  If other changes to the text (other than the correction at Line 32) have been made then they have not been indicated.

Table 1 is not very well laid out.  What does "Serial No." indicate?  The markers for monocyte/macrophage markers has both human and murine surface markers, which are poorly demarcated.  There are column headings that do not indicate significant groupings of information.

The critical issue with this submission is the use of figures that are highly similar to other articles, which is a concern despite the attribution that is given for most of them.

Comments on the Quality of English Language

The writing is satisfactory but still includes some unusual phrasing.

E.g. Ln 20 "Insights into the pathogenetic roles of macrophages in renal disease suggest leveraging the modification of macrophage function as a novel therapeutic and prognosticate approach."

Author Response

Comment number

Comment

Response

1

The authors have significantly increased the length of the Discussion and number of citations. 

We sincerely thank the reviewer for acknowledging our effort

2

The extension to the Discussion includes reference to several relevant studies, although an overarching analysis of these references could be improved.  If other changes to the text (other than the correction at Line 32) have been made then they have not been indicated.

We have made modifications to improve the manuscript and all the changes have been highlighted in yellow. We have attempted to make the review more clinically relevant, highlighting the major experimental studies which paved the path towards the discoveries

3

Table 1 is not very well laid out.  What does "Serial No." indicate?  The markers for monocyte/macrophage markers has both human and murine surface markers, which are poorly demarcated.  There are column headings that do not indicate significant groupings of information

Table 1 and 2 have been modified to make it simpler. The term “serial no” has been removed.

A single flow cytometry surface marker each was used exclusively for human and murine model studies each. To avoid confusion, we have removed the exclusive murine and human markers.

All the markers mentioned in the manuscript in the present version, have been validated both in human as well as in experimental murine models

4

The critical issue with this submission is the use of figures that are highly similar to other articles, which is a concern despite the attribution that is given for most of them.

a. Three figures (figure 2, figure 10 and figure 11 in the earlier manuscript) have been removed from the review.

The figure 2 has been modified removing the immune markers from the table to simplify the content.

Figure 3 is the only image that is reproduced. We are willing to remove the same, as the text in the manuscript conveys all the information. The figure simplifies the complex theoretical component and that is the only image that is reproduced from citation. All the other images are our own.

Two new tables Table 3 and 4 are added to supplement for the removal of earlier Figure 10 and Figure 11.

Comments on the Quality of English Language

The writing is satisfactory but still includes some unusual phrasing.

E.g. Ln 20 "Insights into the pathogenetic roles of macrophages in renal disease suggest leveraging the modification of macrophage function as a novel therapeutic and prognosticate approach."

We thank the reviewer for acknowledging our effort.

We have tried to make further corrections and highlighted the changes.

Reviewer 2 Report

Comments and Suggestions for Authors

This manuscript strives to present a review of the involvement of macrophage lineage cells in human renal disease. Unfortunately, the authors rely heavily of previously published figures rather than developing their own based on the combined information from multiple works. The vast majority of paragraphs do not list references throughout, but rather lump them all at the end. Only in the final section (4. Translational values...) are references properly included for specific findings. Furthermore, references should be including in Tables 1 & 2 as required by the journal format.

Author Response

Comment number

Comment

Response

1

This manuscript strives to present a review of the involvement of macrophage lineage cells in human renal disease. Unfortunately, the authors rely heavily of previously published figures rather than developing their own based on the combined information from multiple works.

a. Three figures (figure 2, figure 10 and figure 11 in the earlier manuscript) have been removed from the review.

The figure 2 has been modified removing the immune markers from the table to simplify the content.

Figure 3 is the only image that is reproduced. We are willing to remove the same, as the text in the manuscript conveys all the information. The figure simplifies the complex theoretical component and that is the only image that is reproduced from citation. All the other images are our own.

Two new tables Table 3 and 4 are added to supplement for the removal of earlier Figure 10 and Figure 11.

2

The vast majority of paragraphs do not list references throughout, but rather lump them all at the end. Only in the final section (4. Translational values...) are references properly included for specific findings.

We have made corrections and ensure relevant citations are listed in the corresponding paragraphs. 

3

Furthermore, references should be including in Tables 1 & 2 as required by the journal format.

References have been included for Tables 1 and 2 as well as the newly added tables 3 and 4

Round 2

Reviewer 2 Report

Comments and Suggestions for Authors

My comments for this version of the manuscript are similar to the last version. References should be included throughout. There are still no references included in Tables 1 and 2. Simply stating that these have been extensively studied in inadequate - people unfamiliar with the field will not know what is or is not accepted in the field. References to original papers are required.

Author Response

The authors sincerely thank the reviewers for their inputs. We have addressed the concerns raised by them and have provided a point by point response. 

Yours sincerely
